# NORMALIZING FLOW BASED METRIC FOR IMAGE GENERATION

## ABSTRACT

We propose two new evaluation metrics to assess realness of generated images based on normalizing flows: a simpler and efficient flow-based likelihood distance (FLD) and a more exact dual-flow based likelihood distance (D-FLD). Because normalizing flows can be used to compute the exact likelihood, the proposed metrics assess how closely generated images align with the distribution of real images from a given domain. This property gives the proposed metrics a few advantages over the widely used Fréchet inception distance (FID) and other recent metrics. Firstly, the proposed metrics need only a few hundred images to stabilize (converge in mean), as opposed to tens of thousands needed for FID, and at least a few thousand for the other metrics. This allows confident evaluation of even small sets of generated images, such as validation batches inside training loops. Secondly, the network used to compute the proposed metric has over an order of magnitude fewer parameters compared to Inception-V3 used to compute FID, making it computationally more efficient. For assessing the realness of generated images in new domains (e.g., x-ray images), ideally these networks should be retrained on real images to model their distinct distributions. Thus, our smaller network will be even more advantageous for new domains. Extensive experiments show that the proposed metrics have the desired monotonic relationships with the extent of image degradation of various kinds.

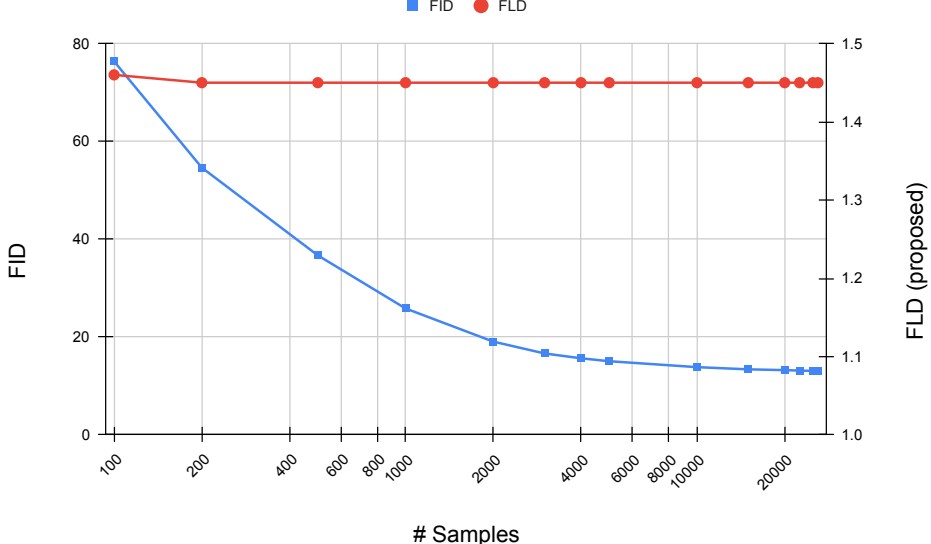

Figure 1: Mean values of FLD and FID by number of generated images clearly demonstrate that FLD bas much better sample efficiency as it achieves reliable results with fewer than 200 samples, whereas FID requires over 20,000 samples to capture a reliable mean score.

# 1 INTRODUCTION

In recent years, several generative adversarial networks (GAN) and diffusion-based image generation models have been proposed for various applications, such as inpainting, text-based image manipulation, and text-to-image generation, that have also caught popular imagination. To compare individual generated images or the overall performance of various models, a robust evaluation metric is essential in determining how closely the generated images resemble real images. This is crucial for understanding whether generative models can produce images that appear realistic to human observers, which is a key goal in many applications.

Evaluating the quality and performance of image generation models presents a unique challenge. Unlike traditional vision tasks such as classification or segmentation, there is no theoretically sound metric to assess performance. This is because generated images require evaluation across multiple dimensions, including quality, aesthetics, realism, and diversity, all of which are deeply rooted in subjective human perception. However, human evaluation is costly and difficult to scale for larger datasets. As a result, researchers rely on automated evaluation techniques to gauge the performance of these models.

In the absence of human evaluation, a good metric of "fakeness" should robustly and monotonically increase with the extents of image degradations of various kinds. Additionally, it should be reliable to assess a given generative model even on a small set of generated images. That is, its mean value should converge with a small number of images generated under similar conditions. Plus, evaluating the metric should be computationally light so that it can be used inside the validation step of training iterations.

Among the various methods proposed for the automatic evaluation of generative models, the Fréchet inception distance (FID) (Heusel et al., 2017) has emerged as the one that is most widely used by practitioners. It approximates the feature distributions of real and generated images as multivariate Gaussians and estimates the distance between them using Fréchet distance (FD). The features used are the activations of the penultimate layer of an Inception-V3 architecture that is pre-trained for classification of the ImageNet dataset. The comparison occurs at object or semantic level, ignoring the fine-grained details of the image. Its simplicity and ease of use have made it a de facto metric in the field despite its known limitations.

Experiments on text-to-image models have highlighted that FID is not an ideal metric as it often disagrees with human ratings, making it unsuitable for evaluating the quality of generated images (Jayasumana et al., 2023). Additionally, statistical tests and empirical evaluation show the limitations of FID in accurately capturing the nuances of image generation (Jayasumana et al., 2023). FID can also be biased depending on the specific model being evaluated. Additionally, it requires a large number of samples to generate a stable and reliable metric value, which can be a limitation in scenarios where fewer samples are available for evaluation or we want to save on computations, such as in a validation step inside a training iteration.

The main goal of generative models is to learn a data distribution that closely approximates the real data, allowing them to generate images with a distribution that mirrors the real one. Ideally, the distance between the two distributions should be minimal, indicating that the generated images are highly similar to real images in terms of their underlying statistical properties.

Our proposed metrics leverage normalizing flows to approximate the data distribution of both generated and real images. By performing density estimation, they assess how well the distribution of generated images aligns with the distribution of real images, providing a more accurate measure of the similarity between the two.

The main contributions of this paper are:

- We introduce a new method for evaluating the performance of generative models by directly assessing the likelihood of each generated image using normalizing flows. To our knowledge, the use of normalizing flows, which can directly and efficiently compute the likelihood of generated images from the real data distribution, has not been previously employed as an evaluation metric.

- We propose a theoretical model that employs dual normalizing flows — one trained on generated images and the other on real images — and demonstrate that this approach is monotonic with respect to various distortions, proving to be an effective evaluation metric.

- We present an efficient and faster single-model normalizing flow as an evaluation metric, showing that it possesses all the essential properties of a good metric. Additionally, it significantly outperforms FID in terms of speed and provides a stable evaluation using a smaller number of generated or real images.

## 2 RELATED WORKS

In this section we summarize previous metrics to evaluate generated images and their limitations, and highlight the relevant properties of normalizing flows on which the proposed metrics are based.

### 2.1 PREVIOUS METRICS AND THEIR LIMITATIONS

Several evaluation metrics have been proposed for generative models, including inception score (IS) Salimans et al. (2016), kernel inception distance (KID) (Bińkowski et al., 2018), FID (Heusel et al., 2017), perceptual path length (Karras et al., 2021), Gaussian Parzen window (Goodfellow et al., 2014), clip maximum mean discrepancy (CMMD) (Jayasumana et al., 2023) as well as human annotation techniques, such as HYPE (Zhou et al., 2019). Among these, inception score and FID have gained significant popularity. The IS utilizes an Inception-V3 model trained on ImageNet-1k to assess the diversity and quality of generated images by analyzing their class probabilities. A key advantage of the Inception Score is that it does not require real images to compute the metric value, making it convenient for certain applications.

KID (Bińkowski et al., 2018) and FID both require the presence of real images alongside the generated images for evaluation. These metrics are computed by measuring the distance between the distributions of real and generated data. KID uses the squared maximum mean discrepancy (MMD) distance, while FID employs the squared FD between the two distributions to assess how well the generated images resemble the real ones.

IS, FID and KID rely on Inception embeddings, which have been trained on 1.3 million images and are limited to the 1,000 classes from the ImageNet-1k dataset (Szegedy et al., 2015). This restriction limits their ability to effectively represent the richer, more complex, and diverse domain images that can emerge from modern image generation tasks, making them less suitable for evaluating certain types of generated content.

Some of the previous works have highlighted the unreliability of evaluation metrics in image generation, especially pointing out the limitations of FID (Chong & Forsyth, 2019). It was shown that FID is a biased estimator and can show significant variation in FID scores for low-level image processing operations, such as compression and resizing (Parmar et al., 2022).

#### 2.1.1 NORMALITY ASSUMPTION

In the computation of FID, the calculation of the FD relies on the assumption that the Inception-V3 embeddings are normally distributed. This is because the closed-form solution for the FD only applies to multivariate normal distributions Dowson & Landau (1982). If the embeddings deviate significantly from this assumption, it can affect the accuracy and reliability of the FID score. Since real image sets typically do not follow normal distributions, this assumption can introduce a significant bias in the computation of the FD and, consequently, the FID score. Additionally, estimating $2048 \times 2048$ dimensional covariance matrices from a small sample of generated and real images, which is required for FID computation, can further exacerbate the error, leading to unreliable evaluations.

This issue is demonstrated by using a 2D isotropic Gaussian distribution at the origin as the reference distribution and measuring the distance between it and a series of mixture-of-Gaussian distributions (Jayasumana et al., 2023), as shown in the Table 1. The second set of distributions is generated as a mixture of four Gaussians, each sharing the same mean and covariance as the reference Gaussian. This example illustrates how the incorrect normality assumption can lead to misleading

Table 1: FD, unbiased FD, and D-FLD values when the normality assumption is violated show the advantage of using D-FLD. The leftmost image represents a reference 2-D Gaussian distribution, progressively from left to right the subsequent distributions deviate further from the true distribution. The FD of all the mixture distributions to the reference distribution, calculated under the normality assumption, remain misleadingly zero (Jayasumana et al., 2023). In contrast, D-FLD accurately captures the increasing deviation from the reference distribution, demonstrating its robustness in scenarios where the normality assumption is violated.

| | | | | | | |
|---|---|---|---|---|---|---|
| FD | 0 | 0 | 0 | 0 | 0 | 0 |
| $FD_\infty$ | 0 | 0 | 0 | 0 | 0 | 0 |
| **D-FLD** | 0 | 0.48 | 1.07 | 2.35 | 6.78 | 11.32 |

distance measurements between real distributions. Since the first mixture shares the same distribution as the reference distribution, we expect any reasonable distance metric to measure zero distance between the two. However, as we progressively move the four components of the mixture distribution further apart from each other, while keeping the overall mean and covariance fixed, the mixture distribution naturally diverges from the reference. Despite this, the FD calculated under the normality assumption, continues to report a misleadingly zero distance, as it assumes normality throughout. Furthermore, the unbiased version of FID, as proposed in previous work (Chong & Forsyth, 2019), also suffers from this limitation since it too relies on the same flawed normality assumption.

MMD has been shown to work better than FD and overcome this limitation, as it does not rely on the assumption that the distributions are multivariate Gaussians. Similarly, as we demonstrate in Table 1, our normalizing flow-based metrics also surpass this limitation. Since they do not require the distributions to be Gaussian, the proposed metrics provide a more accurate and reliable assessment of the difference between real and generated data distributions.

### 2.1.2 IMAGENET PRE-TRAINED MODELS

Previous metrics, e.g., IS, FID, CMMD and KID (Bińkowski et al., 2018), rely on modeling distributions of real images based on features extracted using CNNs that are pre-trained on ImageNet. This creates a bias towards the dataset that was used for pre-training, which limits their applicability to other domains, such as medical x-ray images. To accurately represent images in these specialized fields and new domains, new embeddings must be generated by pre-training on domain-specific datasets. This can often be challenging due to the often limited size of such datasets, making it difficult to produce accurate embeddings.

Hence, it is preferable to have a metric that relies solely on the available real and generated images to create their respective data distributions. This approach allows for a more accurate comparison of how closely the generated images align with the real image distribution. Our proposed normalizing flow-based metrics achieve this by directly comparing the distributions without relying on pre-trained embeddings, making it better suited for diverse and specialized domains.

### 2.2 NORMALIZING FLOWS

Normalizing flows are generative models built on invertible transformations. Among all the generative models that are widely used, such as, GANs, VAEs, and diffusion, normalizing flows are the only type of model for which the likelihood of data can be computed exactly and efficiently (Prince, 2023; Kobyzev et al., 2021). None of the other models explicitly learn the likelihood of the real data. While VAEs model a lower bound of the likelihood, GANs only provide a sampling mechanism for generating new data, without offering a likelihood estimate Kobyzev et al. (2021).

A normalizing flow is able to generate the likelihood of a given sample $\mathbf{x}$ by transforming a complex data distribution into a simpler one, typically a Gaussian distribution, through a series of invertible

transformations. Each transformation maps the input data into a new space while maintaining the ability to compute the log-likelihood of the data $p_{\mathbf{x}}(\mathbf{x})$ in the original space using the change of variables formula:

$$\log p_{\mathbf{x}}(\mathbf{x}) = \log p_{\mathbf{z}}(f(\mathbf{x})) + \log \left| \det \left( \frac{\partial f(\mathbf{x})}{\partial \mathbf{x}} \right) \right|, \tag{1}$$

where $f(\mathbf{x})$ is the transformation applied by the normalizing flow, $p_{\mathbf{z}}(f(\mathbf{x}))$ is the likelihood of the transformed sample in the latent space (e.g., the probability density of $\mathbf{z}$ under a Gaussian distribution), and $\det \left( \frac{\partial f(\mathbf{x})}{\partial \mathbf{x}} \right)$ is the Jacobian determinant of the transformation that accounts for the change in volume during the transformation (Kobyzev et al., 2021).

The function $f(\mathbf{x})$, which represents the transformation applied by the normalizing flow, plays a crucial role in mapping the complex data distribution to a simpler latent distribution. This transformation can be modeled by neural networks layers, but the layers so used must be an invertible function to ensure that the mapping between the original data and the latent space can be reversed. Additionally, $f(\mathbf{x})$ needs to have a tractable Jacobian determinant to facilitate efficient computation of the likelihood. To learn a data distribution, we train the parameters of the neural network $f(\mathbf{x})$ to maximize the likelihood of training data (Prince, 2023).

## 3 THE FLOW-BASED LIKELIHOOD DISTANCE METRIC

We propose two evaluation metrics for generative models using the likelihood estimates that can be obtained from normalizing flows. We first take a set of real images and a set of generated images.

$$\mathcal{R} = \{r_1, r_2, \ldots, r_n\} \qquad \text{(set of real images)}$$

$$\mathcal{G} = \{g_1, g_2, \ldots, g_m\} \quad \text{(set of generated images)}$$

### 3.1 DUAL FLOW-BASED LIKELIHOOD DISTANCE

For calculating the dual-flow based likelihood distance (D-FLD), we first use two separate normalizing flows $N_r$ and $N_g$ and train them using real images $\mathcal{R}$ and generated images $\mathcal{G}$, respectively. This enables the flow $N_r$ to learn the probability distributions of the real images and flow $N_g$ to learn the distribution of generated images. Once we have the trained $N_r$ and $N_s$, we then pass all the images, both real and generated through both the flow and evaluate the log-likelihood of each image. For all images $x$ in sets R and G.

$$\mathcal{L}_r = N_r(x) \qquad \forall x \in \mathcal{R} \cup \mathcal{G} \tag{2}$$

$$\mathcal{L}_g = N_g(x) \qquad \forall x \in \mathcal{R} \cup \mathcal{G} \tag{3}$$

This give the log-likelihood of the image with respect to the real distribution ($\mathcal{L}_r$) and the log-likelihood of the image with respect to the generated distribution ($\mathcal{L}_g$). If the generated distribution closely approximates the real distribution, the likelihood of each image with respect to the two sets $\mathcal{R}$ and $\mathcal{G}$ should be similar and their difference should be small. We then compute the distance $d$ for an image as the absolute value of difference in the likelihoods obtained from $N_r$ and $N_g$. For a dataset, we compute the average $d$ value for all images in $\mathcal{R}$ and $\mathcal{G}$ as our metric, $m$. We take its log value to make its range more interpretable.

$$d(x) = |\mathcal{L}_r - \mathcal{L}_g| \qquad \forall x \in \mathcal{R} \cup \mathcal{G} \tag{4}$$

$$m = \frac{\sum_{x \in \mathcal{R} \cup \mathcal{G}} d(x)}{|\mathcal{R}| + |\mathcal{G}|} \tag{5}$$

$$\text{Dual-FLD} = \log_2(1 + m) \tag{6}$$

If the real and generated images come from similar distributions, then the two flows will also model similar distributions and the $d$ values will be small. The more the generated distribution moves away from the real distribution, the $d$ value for an image will also increase. This is because real images

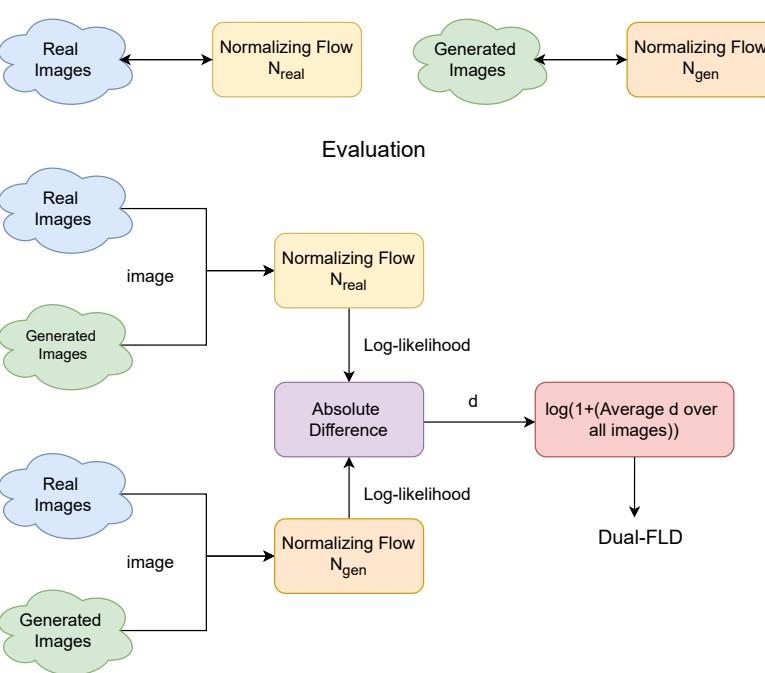

Figure 2: The process of computing D-FLD using two normalizing flows. In the training phase, two normalizing flows are independently trained on real and generated images. In the evaluation phase, each image is passed through both flows, and the absolute difference between the log-likelihoods of both flow is computed, averaged over all images and then transformed to produce the final metric.

will be given higher likelihood by the $N_r$ but will be given lower likelihood by $N_g$ and vise versa for generated images. When we compute the average value of $d$ across all the images, it gives a sense of how closely the generated distribution approximates the real one. Hence, the D-FLD score represents the distance between the two distributions. The training and computation of D-FLD is shown in the Figure 2.

### 3.2 FLOW-BASED LIKELIHOOD DISTANCE

The D-FLD requires us to train two normalizing flows – one each on real and generated data – and to pass all the images through both the flows. This consumes time and computational resources. Therefore, we propose a simpler and more practical metric which only needs training of a single normalizing flow with just the real data $\mathcal{R}$. This metric can use a pre-trained network.

We first train a single normalizing flow $N$ using only the real images in $\mathcal{R}$. We then evaluate the average log-likelihood for all images in the real set $\mathcal{R}$ using $N$. We also evaluate the average log-likelihood of all the images in generated image set $\mathcal{G}$ using $N$. The FLD score is computed as the ratio of average log-likelihood of real samples by that of generated samples as shown in Figure 3.

$$\text{FLD} = \frac{\dfrac{\sum_{x \in \mathcal{R}} \mathcal{L}_r(x)}{|\mathcal{R}|}}{\dfrac{\sum_{x \in \mathcal{G}} \mathcal{L}_r(x)}{|\mathcal{G}|}} \tag{7}$$

Since the normalizing flow $N$ is trained only on the real images $\mathcal{R}$, passing the real dataset $\mathcal{R}$ through the normalizing flow will result in very high log-likelihood values for the each image in it. In contrast, when we pass the generated images through the normalizing flow $N$, if the generated

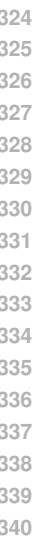
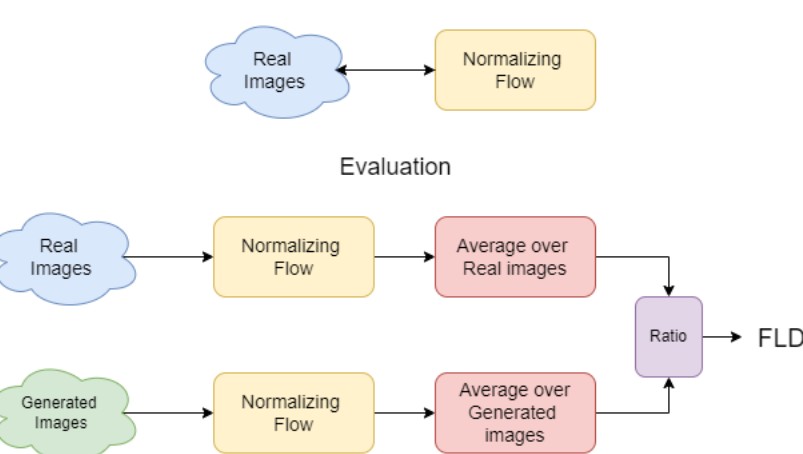

Figure 3: The process of computing FLD using a single normalizing flow. In the training phase, a normalizing flow is trained on real images. In the evaluation phase, real and generated images are passed through the flow, and their log-likelihoods are computed and averaged separately and their ratio is the final metric.

data distribution closely resembles the real data distribution or if the generated images are very similar to the real ones, they would also yield a high average log-likelihood value, similar to that of the real images. As a result, the ratio of the two values will approach one. However, if the generated data distribution deviates significantly from the real distribution, the normalizing flow will assign much lower log-likelihood values to the generated images, causing the average log-likelihood for the generated samples to decrease and the metric to increase much above 1. A graphical illustration of these concepts is provided in Figure 9 and Figure 10 in the Appendix.

## 4 EXPERIMENTS AND RESULTS

We now describe the datasets, implementation details, experimental design, and the results.

### 4.1 DATASETS AND IMPLEMENTATION DETAILS

We used CIFAR-10 (Krizhevsky, 2009) and CelebA-HQ (Karras et al., 2018) datasets for our experiments to analyse the behaviour of our proposed metric and compare it with FID. We used CIFAR-10 at $32 \times 32$ and CelebA-HQ at $256 \times 256$ resolution. All experiments were run on Nvidia A6000 GPU.

We used a simple normalizing flow (Lippe, 2024) for calculating D-FLD which has four variational dequantization layers (Ho et al., 2019) followed by eight coupling layers (Dinh et al., 2017). For computing FLD, we use a multi-scale normalizing flow with four variational dequantization layers (Ho et al., 2019) followed by eight coupling layers (Dinh et al., 2017). We apply checkerboard masks throughout the networks (Dinh et al., 2017). More details are provided in the Appendix.

### 4.2 EVALUATING IMAGE DISTORTIONS

We tested if FLD and D-FLD monotonically increased with levels of various types of image distortions, and found the trends to be robust and as desired. Additionally, when a small amount of Gaussian noise was added to the images, we observed that FID does not give a monotonically increasing trend and incorrectly evaluates noisy images as having better quality than those with lower noise. On the other hand, D-FLD has a monotonic trend, as seen in Figure 4. This allows accurate characterization of the images as progressively noisier, and suggests that D-FLD is a more

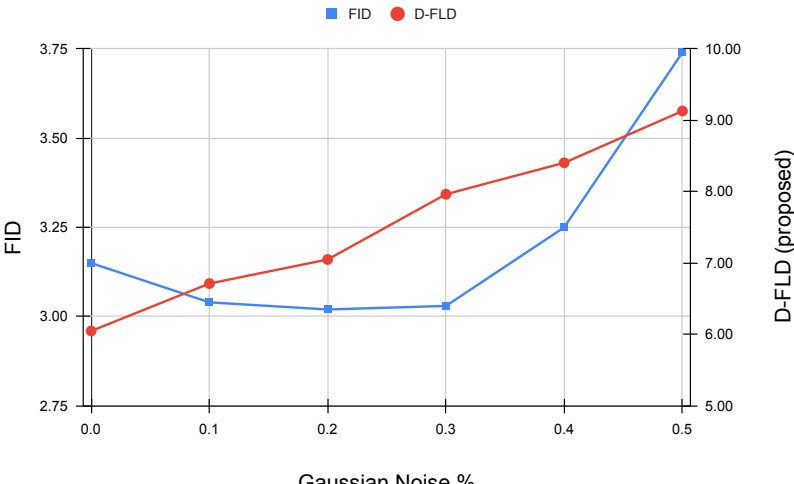

Figure 4: Monotonicity and robustness of proposed metric: Adding various levels of Gaussian noise to CIFAR-10 images show that FID does not show a monotonic relationship with the proportion of noise added, while the proposed metric D-FLD shows a monotonic trend.

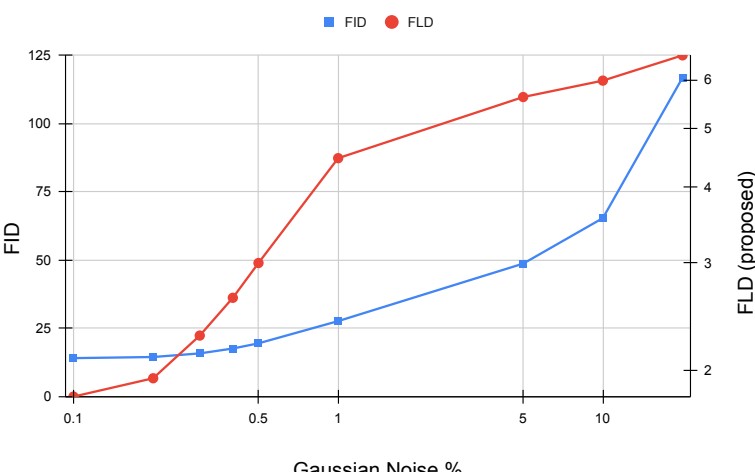

Figure 5: Behavior of FLD and FID as Gaussian noise is added to CelebA-HQ images. While both metrics increase monotonically, FLD demonstrates greater sensitivity in the lower noise range compared to FID. Since state-of-the-art generative models require discrimination of subtle levels of noise, this heightened sensitivity of FLD offers a more precise evaluation of advanced generative models.

reliable and effective metric than FID. Our results matches previous work which have shown that FID sometimes reverses in trend undesirably when subtle distortions are applied Jayasumana et al. (2023).

Figure 5 illustrates the behavior of FLD and FID when Gaussian noise is added to CelebA-HQ images. It is evident that FID is more sensitive to larger amounts of noise, as the variation in its values becomes significant only when a substantial level of noise is introduced. In contrast, FLD is highly sensitive even in the low noise range. When only 0.1% to 0.5% Gaussian noise is added, the variation in FID is minimal, whereas our FLD metric shows a much greater sensitivity. Given

that modern generative models excel at producing high-quality images, it is crucial for evaluation metrics to detect subtle nuances and small amounts of noise with precision. FID fails to meet these demands from the latest models, but FLD is able to do so effectively. Therefore, FLD is a far more suitable metric for evaluating the performance of state-of-the-art generative models.

The performance of FLD, D-FLD and FID on other kinds of distortion, such as Gaussian blur and salt and pepper noise, for CelebA-HQ and CIFAR-10 datasets are shown in the Appendix.

## 4.3 PROGRESSIVE IMAGE GENERATION

Modern image generation models, such as diffusion, iteratively generate high quality image by refining noisy images in steps. It was shown that FID does not behave monotonically when denoising iterations proceed (Jayasumana et al., 2023) and hence should not be used as a metric to evaluate these models, especially for the final iterations when the noise level is low. Figure 6 and Figure 7 shows the FLD values for the final iterations in a 1000 step diffusion process (Ho et al., 2020) and shows that it can accurately capture the differences in image quality.

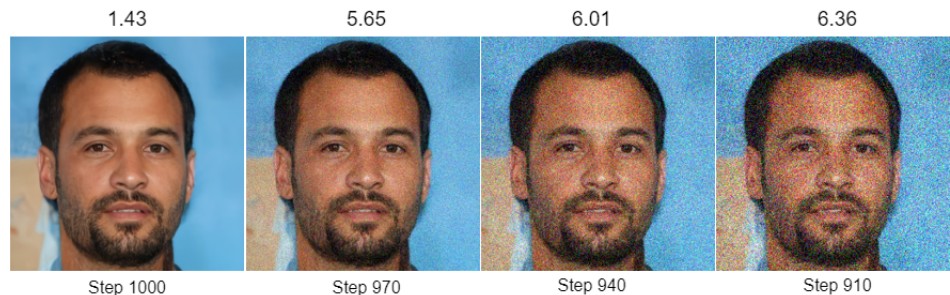

Figure 6: The behavior of FLD in diffusion models. The FLD values are shown at the top, while the number of diffusion steps is displayed at the bottom. As the number of diffusion steps increases, the image quality improves, which is captured by the decreasing FLD values, as desired.

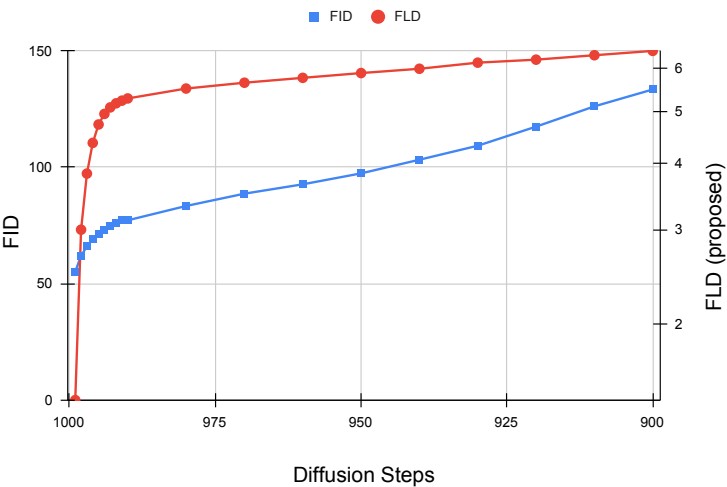

Figure 7: The behavior of FLD and FID in denoising diffusion probabilistic model with 1000 steps. FLD is strictly monotonic and accurately captures the denoising process.

## 4.4 EFFICIENCY

Earlier work (Chong & Forsyth, 2019; Jayasumana et al., 2023), along with our own experiments, has demonstrated that computing the FID value requires a large number of images to get a stable mean score, which translates to a poor sample efficiency. Our experiments indicate that more than 20,000 images are necessary to reliably estimate FID, and using smaller sample sizes results in an unreliable estimate, which can significantly deviate from the true value, as shown Figure 1. Even the metrics that were proposed after FID, such as CMMD, require at least a few thousand samples for a reliable estimation of a generative model's average behavior (Jayasumana et al., 2023). Figure 1 clearly demonstrates that FLD metric achieves high sampling efficiency, as it provides a stable and reliable metric with just 200 samples. This showcases the robustness and efficiency of FLD compared to other metrics, and makes it attractive even inside training loops, such as during validation steps.

The model used for normalizing flow in FLD also uses an order of magnitude fewer parameters (1.9 M) compared to the Inception-V3 model (23.8 M parameters) used in FID computation. Thus, FLD can also be trained on edge devices. These advantages makes FLD much more efficient for evaluation of generative models.

Just like FID, FLD needs to be trained only once on the real images of a given domain. Once the normalizing flow is trained, evaluation of generated images are fast since we do not need to retrain the normalizing flow.

## CONCLUSION

In this paper, we have demonstrated that FID is not a reliable metric for evaluating generative models. Our findings highlight several critical issues with FID, including its requirement of a large number of image samples to provide a reliable estimate, which limits its efficiency and applicability in real-time or low-sample scenarios. Additionally, FID exhibits non-monotonic behavior, particularly when dealing with diffusion models and image distortions. To address these limitations, we have introduced two new metrics based on normalizing flows: dual-FLD and FLD. These metrics have proven to be monotonic with respect to image distortions and perform well with diffusion models. Furthermore, our metrics are highly sample efficient, requiring only a few hundred images to provide a reliable estimate. This makes them a more practical and effective alternative to FID for evaluating the quality of images generated by modern generative models.

As a limitation, we advise practitioners to use our metrics with caution for evaluating the performance of other generative models, especially those based on normalizing flows. This is because we have tested the metric only on images generated by adding distortions to real images and on those generated by diffusion-based models. Additionally, due to limitations in computational resources, we were unable to test our metrics on larger generative models, such as text-to-image models, or on more extensive datasets with diverse categories and domains. These remain areas for future exploration and evaluation.

We believe that our work will spark further research in the search for better and more efficient evaluation metrics for generative models. Additionally, there is potential to optimize normalizing flow-based evaluation metrics by constructing even more efficient and expressive normalizing flows or training on more advanced normalizing flows than those we have used in this study. This opens the door for continued improvement in the field of generative model evaluation.

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

# A APPENDIX

## A.1 SAMPLING EXPERIMENT

To determine the number of samples required for the FID and FLD to reach reliable values, we conducted an experiment using a pre-trained Denoising Diffusion Probabilistic Model (DDPM) (Ho et al., 2020) trained on the CelebA-HQ dataset (Chen, 2023). We generated a set of images from this model and systematically varied the number of images sampled from this set to compute the FID and FLD metrics. By measuring these metrics across different sample sizes, we aimed to assess the influence of sample quantity on the convergence behavior of FID and FLD. For each sample size, we ran the experiment 10 times with different images sampled randomly from both real and generated image sets.

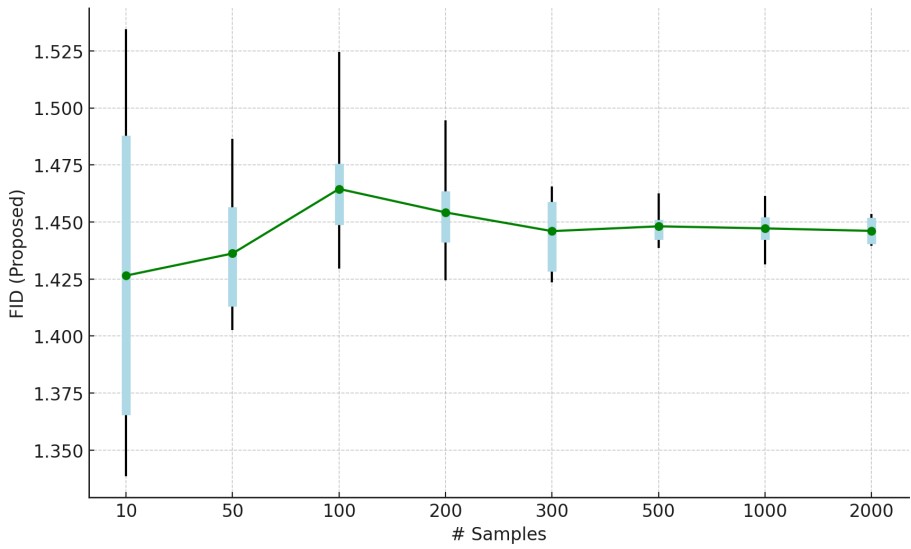

Figure 8: The behavior of FLD across different sample sizes clearly demonstrates that FLD achieves reliable results with fewer than 200 samples. The variance in FLD computed clearly goes down when sample size is more than 200. The mean FLD (green) is reported after 10 runs for each sample size.

## A.2 RESULTS FOR IMAGE DISTORTIONS

**Gaussian Noise:** We construct a noise matrix $N$ with values drawn from a $\mathcal{N}(0, 1)$ Gaussian distribution and scaled to the range $[0, 255]$. The noisy image is then computed by combining the original image matrix $X$ with the noise matrix $N$ as $(1-\alpha)\cdot X + \alpha\cdot N$, where $\alpha \in \{0, 0.001, 0.005, 0.01, etc\}$ determines the amount of noise added to the image. A larger value of $\alpha$ introduces more noise and the noisy image is clipped to ensure that all pixel values remain within the valid range $[0, 255]$.

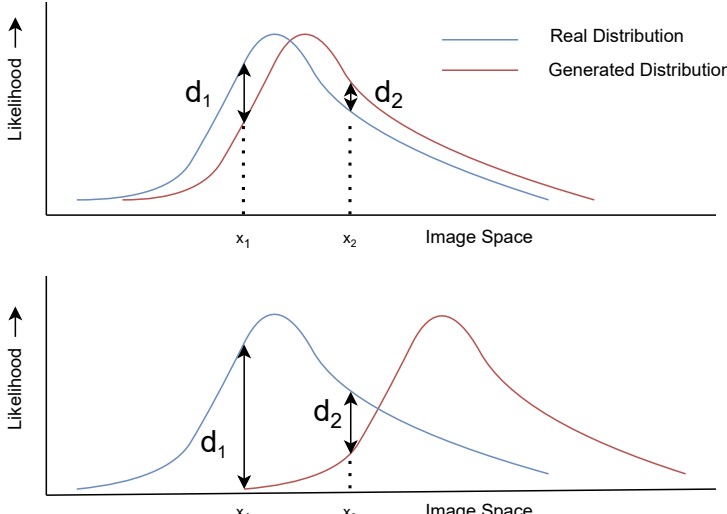

Figure 9: Visual interpretation of D-FLD metric: The figure visualizes two sets of probability density functions estimated using normalizing flows. The blue curve represents the distribution of the normalizing flow trained on real data, while the red curve represents the distribution trained on generated data. In the top graph, the distributions are relatively similar, resulting in smaller differences in likelihood values $d_1$ and $d_2$ for two images $x_1$ and $x_2$ respectively. Thus, lower difference in likelihood values indicates a closer match between the real and generated data distributions, especially when averaged over the entire image space. In contrast, the bottom graph shows more dissimilar distributions, leading to larger difference $d_1$ and $d_2$ in likelihood values, demonstrating that as the dissimilarity between the distributions increases, the average likelihood difference across the image space also increases. This highlights how closely aligned probability densities lead to smaller likelihood differences, while greater divergence results in larger likelihood differences across the image space. Thus, difference in likelihood can be used as a good metric for evaluating the distance between two data distributions.

**Gaussian Blur:** The image is convolved with a Gaussian kernel with standard deviation specified by a blur readius. The larger the values of blur radius $r$ greater the amount of blur. In this case, $r \in \{0, 0.25, 0.5, 1, 2\}$, and the Gaussian blur is applied uniformly across the image.

**Salt and Pepper Noise Addition:** To add salt and pepper noise to an image, random values are generated for each pixel. The probability $p$ controls the proportion of pixels that will be altered. For salt noise, pixels with random value less than $\frac{p}{2}$ are set to the maximum intensity value (255). Similarly, for pepper noise, pixels with random value greater than $1 - \frac{p}{2}$ are set to the minimum intensity value (0). The amount of noise is directly proportional to the value of $p$, here $p \in \{0, 0.001, 0.005, 0.01, etc\}$ with larger values of $p$ resulting in a higher number of noisy pixels in the image.

## A.3 IMPLEMENTATION DETAILS

For computing FLD, we used a multiscale flow-based generative model for images that integrates variational dequantization with a sequence of coupling layers parameterized by gated convolutional networks (Lippe, 2024). The architecture comprises 12 coupling layers: 4 within the Variational Dequantization Layer (with hidden channels of 16, using checkerboard masks), 2 coupling layers after dequantization (hidden channels of 32, using checkerboard masks), 2 coupling layers after the first squeeze flow layer (with input channels of 12 and hidden channels of 48, employing channel-wise masks), and 4 coupling layers in the final block after the second squeeze flow (with input channels of 24 and hidden channels of 64, also using channel-wise masks). The model also includes 2 squeeze flow layers to adjust spatial and channel dimensions and 1 split flow layer to partition the data for efficient modeling. This design enables efficient learning of complex image distributions across mul-

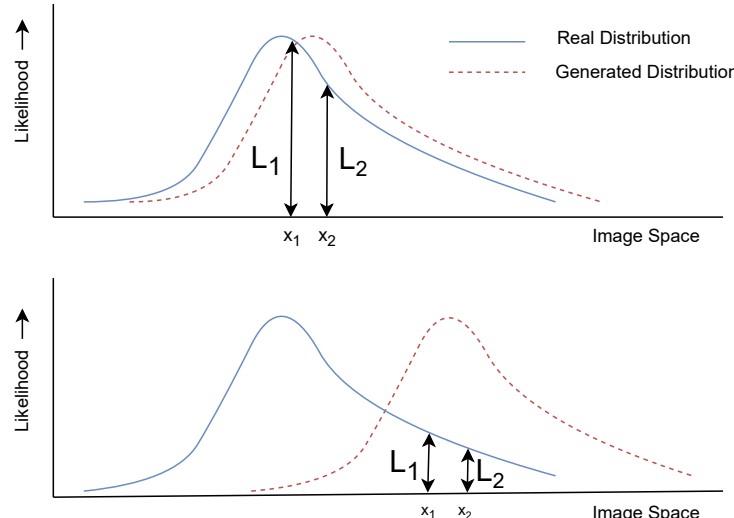

Figure 10: Visual interpretation of FLD metric: The figure illustrates the real image data distribution estimated by a normalizing flow (shown in blue), and the generated image distribution (shown as dotted red lines), which is not modeled by a normalizing flow. When we take generated images, most of them will belong to regions with high likelihood in the generated distribution, as indicated by points $x_1$ and $x_2$. In the top figure, we demonstrate a scenario where the real data distribution and the generated distribution are closely aligned and similar. We see that the likelihood of the generated images with respect to the real data distribution, $L_1$ and $L_2$ will be very high, emphasizing the strong similarity between the distributions. When we average over all the generated images, the average likelihood value will also be higher. Similarly, in the bottom image, when the real distribution and the generated distribution are dissimilar or further apart, the likelihood of the generated images with respect to the real distribution is significantly lower. As we evaluate and average the likelihood over all generated images, this scenario results in a much lower overall likelihood value compared to the case where the real and generated distributions are closely aligned, as depicted in the top image. This highlights the increased distance between the distributions and its impact on the likelihood evaluation. Hence, the likelihood of generated images with respect to real data distribution can be used as a good metric for evaluating the distance between two data distributions.

tiple scales while maintaining invertibility for exact likelihood computation and facilitating efficient sampling.

The split flow layer is used to divide the input into two parts during the forward pass. One part is passed directly through the flow, while the other part is evaluated against a Gaussian prior distribution. This approach reduces the dimensionality of the transformed data, allowing the model to focus on lower-dimensional latent representations in later layers.

For experiments involving sampling efficiency, we used a CelebA-HQ pre-trained diffusion model for generating images (Chen, 2023). For experiments on checking the performance of our metric on progressive image generation, we used the original DDPM model (Ho et al., 2020).

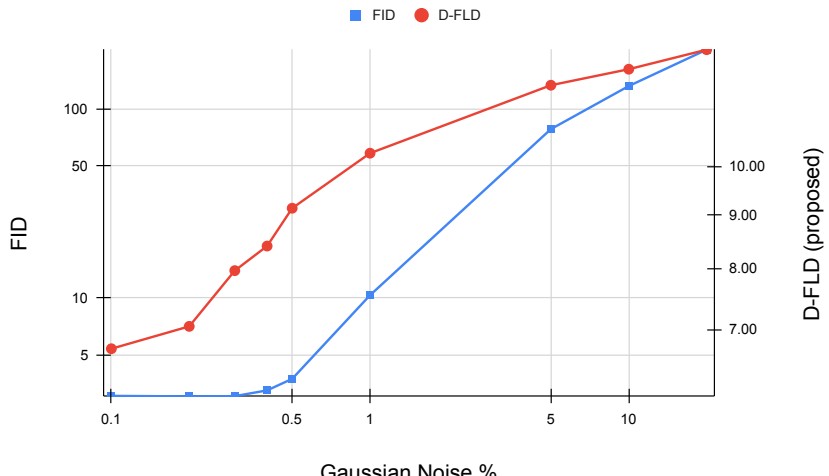

Figure 11: The behavior of D-FLD and FID as Gaussian noise is added to CIFAR-10 images. This is an extension of Figure 4 with higher noise added.

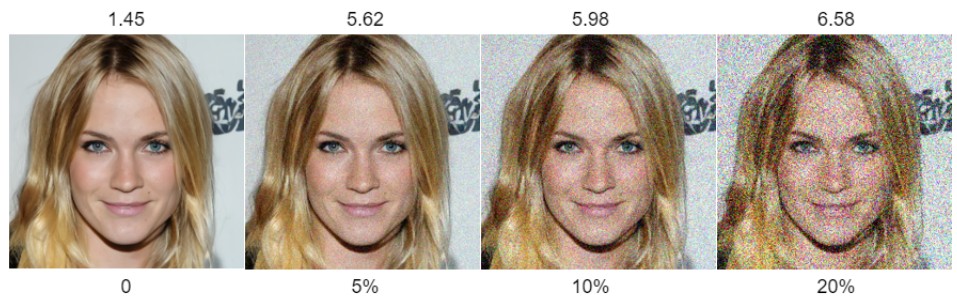

Figure 12: The behavior of FLD when Gaussian noise is added to CelebA-HQ images. The FLD values are shown in the top row and the percentage of noise added ($\alpha$) is shown in the bottom row.

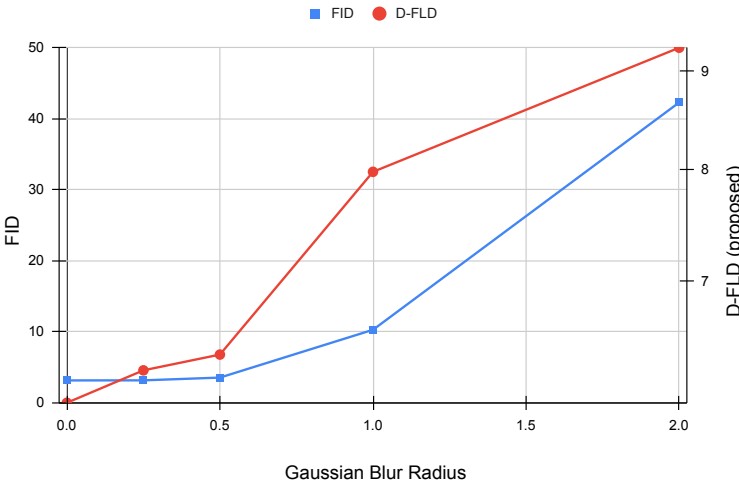

Figure 13: The behavior of D-FLD and FID as Gaussian blur is added to CIFAR-10 images.

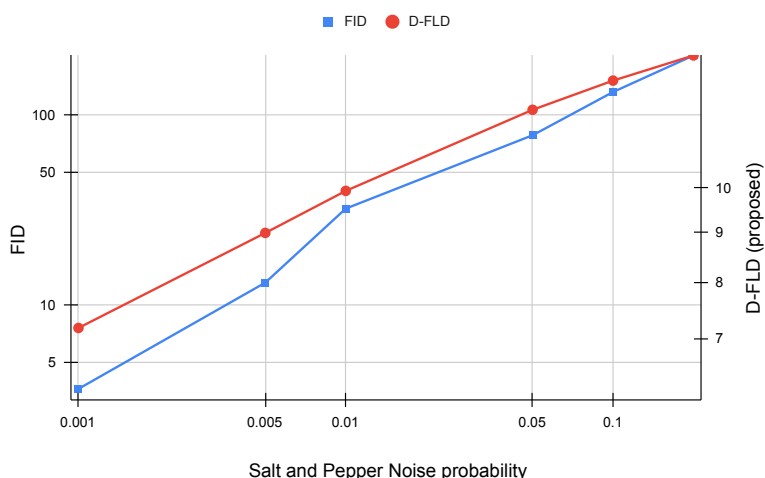

Figure 14: The behavior of D-FLD and FID as salt and pepper noise is added to CIFAR-10 images.

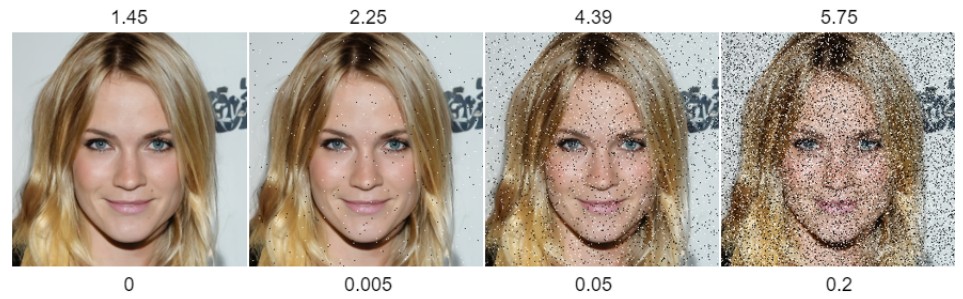

Figure 15: The behavior of FLD when salt and pepper noise is added to CelebA-HQ images. The FLD values are shown in the top row and the probability controlling the noise added ($p$) is shown in the bottom row.

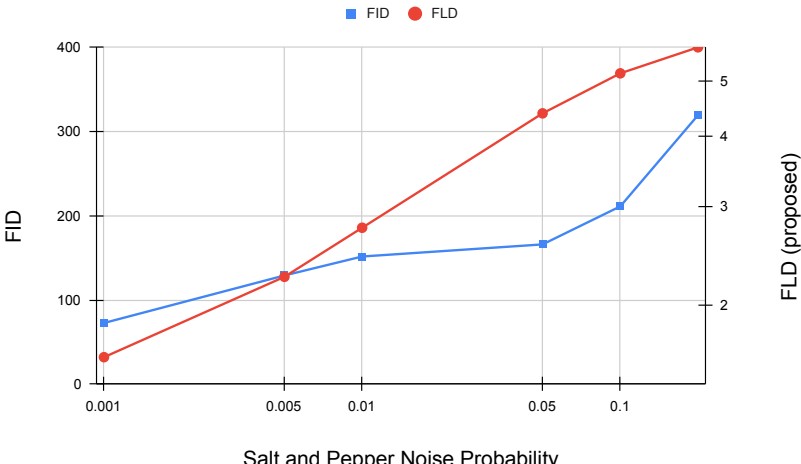

Figure 16: The behavior of FLD and FID as salt and pepper noise is added to CelebA-HQ images.

