# OpenReview forum: "Normalizing Flow Based Evaluation Metrics for Image Generation"
_ICLR.cc/2025/Conference — Submitted to ICLR 2025_

### Official Review · Reviewer_6rkA · 2024-11-04

**Soundness:** 1
**Presentation:** 1
**Contribution:** 1
**Rating:** 1
**Confidence:** 3

**Summary:**

This paper introduces two novel metrics based on normalizing flows to evaluate the realism of generated images more accurately and efficiently than existing methods, especially Frechet Inception Distance (FID). The proposed metrics: Flow-based Likelihood Distance (FLD) and Dual-Flow Likelihood Distance (D-FLD), use exact likelihood estimation to capture the similarity between real and generated image distributions. Unlike FID, these metrics achieve stable results with far fewer images, making them computationally lighter and more suited to new image domains.

**Strengths:**

1) The article addresses an interesting and topical problem that is of practical relevance - in many applications the FID metric may be inadequate, indicating a real need for the development of new metrics to assess the quality of generated images.
2) The use of the Normalizing Flow model is an important step toward overcoming the limitations imposed by the normal distribution assumption typical of FID. Thus, the proposed metric allows a more precise assessment of the quality of the generated data, even if the distributions do not meet the normality assumption.
3) Additionally, the authors have taken into account the flexibility aspect of the measure - the proposed metric works effectively for different counts of real ($n$) and generated ($m$) image sets.

**Weaknesses:**

1) No extensive comparison between the proposed and existing metrics leveraging various generative models. The proposed metrics were tested only on images generated by adding distortions to real images and on samples produced by the diffusion-based model. It's definitely not enough and at least a couple of real models comparison is required.
2) The authors suggest (Section 4.2) that the high sensitivity of the FLD metric to small levels of noise could be an advantage in the context of training generative, high-quality models. Unfortunately, no experiments have been conducted to confirm this hypothesis, which reduces the credibility of this statement.
3) The claim that limitations on large datasets are due to computational constraints (in the conclusion) conflicts with section 4.4, where the low parameter count for normalizing flows is described as being compatible with training on edge devices.



Figures:
a) Figure 1 does not specify the dataset used, which limits reproducibility.
b) Inconsistency in X-axis scales: Figure 4 uses a linear scale, while Figure 5 uses a logarithmic one.
c)  Applying a logarithmic transformation seems overly simplistic, and it’s hard to ensure a fair comparison with FID in Figure 4. FLD comparisons should be included.
d) The authors' presentation of the diffusion model results does not follow common practice. The authors refer to the steps made by the diffusion model (x_1000) instead of the timestamps (x_0 is a real data sample). Figures 6 and 7 can confuse the reader.

**Questions:**

Q1: While the authors mention that the IS, FID, and KID metrics are limited to 1000 classes from the ImageNet-1k dataset, no experiments have been performed to evaluate the performance of the proposed metric on this dataset. Can the authors provide results of the performance of their metric on ImageNet-1k to allow comparison with widely used image generation quality metrics?
Q2: The FLD and D-FLD metrics are based on training the Normalizing Flow model on a given dataset. Can the authors clarify whether and to what extent the process of training this model (e.g. the tuning of hyperparameters) is dependent on the specifics of the dataset? If such differences exist, can the authors provide recommendations for parameter selection to improve the application of the metric under different conditions?
Q3: The authors note that the metric stabilizes at around 200 samples, it is unclear what size dataset is required to effectively train the Normalizing Flow model to ensure consistency of results. Could the authors specify the minimum dataset size necessary to achieve meaningful results?
Q4: There is a lack of comparison of commutation times for FID and FLD metrics. Given the amount of work required to train the Flow models, there is an assumption that the calculation of the D-FLD metric may be time-consuming. It would be interesting to see a numerical comparison of the calculation times of the two metrics, which would allow a better assessment of their practicality.

---

### Official Review · Reviewer_W3vU · 2024-11-04

**Soundness:** 3
**Presentation:** 3
**Contribution:** 3
**Rating:** 5
**Confidence:** 4

**Summary:**

This paper proposes an efficient, likelihood-based metric to evaluate generated samples. The authors suggest training a flow-based model on the training set and using the trained model to estimate the likelihood distance between training and generated images. This method stabilizes faster than FID and is parameter-efficient, as it requires training smaller models than FID-based approaches.

**Strengths:**

- The metric stabilizes rapidly, requiring only 100-200 samples.
- The flow-based likelihood distance is straightforward and easy to interpret.
- The method is more sensitive to image distortions.
- The paper is well-structured and easy to follow.

**Weaknesses:**

1. Reliability of Likelihood as a Metric: The suitability of likelihood as a metric is debatable. As noted in [1], likelihood may not be reliable for deep generative models, which often lack calibration. This paper particularly studies flow-based models, which have similar limitations. Additionally, [2] suggests that likelihood and visual fidelity are largely independent in high-dimensional data. Unlike FID, which computes the distance between real and generated samples in feature space and aligns more closely with human perception [3], it’s unclear if the proposed FLD metric can serve as a reliable FID replacement. To substantiate this claim, the authors should demonstrate that the proposed method is well-calibrated and capable of out-of-domain detection. Following [1], it would be beneficial to analyze the data distribution and model curvature to show that FLD is reliable.
2. Consistency: References [2] and [3] indicate that FID and likelihood are largely independent. Nevertheless, likelihood could still correlate with visual fidelity. Comparing FID and FLD across different models might reveal if these metrics are somewhat consistent and correlated. Additionally, more experiments are needed to demonstrate FLD's reliability and clarify if a better FLD score truly indicates higher quality in generated samples.
3. Parameter Efficiency and Scalability: The paper claims parameter efficiency by comparing FLD to Inception-V3, which is pre-trained on ImageNet. To support this claim, the authors should demonstrate that the 1.9M flow model can sufficiently represent ImageNet and produce convincing evaluations. It is uncertain if this 1.9M flow model can reliably model ImageNet1K, or if the 23.8M Inception-V3 is over-parameterized for CIFAR-10, making the parameter comparison potentially unfair. Additionally, scalability might pose an issue if this metric is extended to ImageNet or even larger datasets like Laion5B. To maintain consistency and reliability on such scales, the authors should clarify the model size and time required.
4. Convergence: Figure 1 suggests that FLD converges more quickly than FID. However, in practice, model comparison during training is based on relative, not absolute, metric values. For FID, we might not actually need 20k samples. Appendix A.1 provides FLD's variance and mean, but it would be informative to include variance bars for FID and compare the relative variance (variance/mean) between the two metrics.
5. Robustness: The paper asserts that FLD is sensitive to image distortions. However, in Figure 4, Gaussian noise is applied at a level of 0.1, raising questions about the metric's robustness. If a small amount of noise significantly impacts FLD, it might be susceptible to adversarial noise. The practical use cases for such sensitivity require further clarification.
6. Usefulness for Probabilistic Models: While FLD may offer a likelihood evaluation method for non-probabilistic generative models, both diffusion models and normalizing flows allow for relatively precise likelihood computation without requiring additional training. Although the 1.9M model may be faster, it is less precise than diffusion models. Clarifying the degree of precision loss in FLD when using a parameter-efficient model would strengthen this argument.
[1] Nalisnick, E. et al. (2019). Do deep generative models know what they don’t know? International Conference on Learning Representations (ICLR). Available at https://arxiv.org/pdf/1810.09136.
[2] Theis, L. et al. (2016). A note on the evaluation of generative models. International Conference on Learning Representations (ICLR). Available at https://arxiv.org/pdf/1511.01844.
[3] Heusel, M. et al. (2017). GANs trained by a two time-scale update rule converge to a local Nash equilibrium. Advances in Neural Information Processing Systems (NeurIPS). Available at https://arxiv.org/abs/1706.08500.

**Questions:**

Please check the weaknesses

---

### Official Review · Reviewer_X6ax · 2024-11-07

**Soundness:** 2
**Presentation:** 2
**Contribution:** 2
**Rating:** 3
**Confidence:** 4

**Summary:**

The paper introduces two new metrics for evaluating generated images, based on normalizing flows. The first metric (FLD) is simpler and uses a single normalizing flow trained on real images. The second metric (D-FLD) uses two normalizing flows trained on real and generated images. By empirical study the authors demonstrate the advantages of their metrics over FID, mainly in terms of sample efficiency and sensitivity to subtle image quality differences.

**Strengths:**

1.	The paper addresses an important problem – how to evaluate the quality of images generated by generative models.
2.	The authors show some evidence that the method is superior to the FID metric in terms of sample efficiency and sensitivity to subtle image quality differences.
3.	The paper is easy to read.

**Weaknesses:**

I think the empirical part in the paper is rather weak to show the robustness and usefulness of the proposed metrics. In addition, the comparison to prior work is very limited.

Specifically:

1.	The authors test only one type of generative model (DDPM) and only two (rather small) datasets (CIFAR-10 and CelebA-HQ). I think the types of distortions in real generative models can be different from the synthetic distortions that considered in the paper, and I think the authors should check how the new metrics capture real distortions.
2.	The authors compare only to the FID metric. I think the authors should compare in the experiments to other metrics like CMMD (which is shown to outperform FID, e.g. in sensitivity to small distortions). Also, I think it will be useful to add a table comparing the properties of different metrics. In addition, the authors propose 2 metrics, but don’t compare between them in the experiments (on the same dataset).
3.	I think the authors should compare the computational complexity (time) of training a model like Inception for FID and training normalizing flows for their metrics. Also the computational cost for computing each metric. The authors claim that their metrics “outperform FID in terms of speed” but there is no evidence for that in the paper.
4.	The authors should state how many real and generated images were used in all experiments (e.g. Figures 4,5).

Additional comments:

-	In figure 8 I guess the y-axis label should be “FLD”. I recommend to add also the second metric, and also FID to this plot.
-	I recommend to add the second metric (D-FLD) to Figure 1.
-	In Figure 7 it is unclear why FID is not monotonic. Are there 2 points with exactly the same FID value ?
-	I think the paper would benefit from a formal (mathematical) introduction of other metrics like FID.


Overall, I think the paper is not ready for publication at the current stage.

**Questions:**

How sensitive the new metrics to the architecture used for normalizing flows ?

---

### Official Review · Reviewer_4nMi · 2024-11-09

**Soundness:** 3
**Presentation:** 4
**Contribution:** 3
**Rating:** 6
**Confidence:** 2

**Summary:**

This paper proposes a new image generation evaluation metric, namely FLD and D-FLD. It uses normalizing flows to assess the likelihood and how closely generated images align with the distribution of real images from a given domain. Compared to FID, the proposed FLD are (1) more sample efficient, making it more ffriendly to small-sample domain like X-ray images; (2) more computationally efficient; (3) more stable and monotonic when adding Gaussian noises.

**Strengths:**

The idea of proposing a new image generation metric makes intuitive sense. It has long been shown the popular FID metirc has certain limits. This work leverages the likelihood assessment ability of normalizing flows in a clever way, and shows the empirical superiority of FLD on CIFAR-10 and CelebA-HQ.

**Weaknesses:**

See questions.

**Questions:**

1. Looks like FLD still needs a pre-trained network. In this paper, I assume the normalizing flow is trained on CIFAR-10 and CelebA-HQ? If so, does the model need to be retrained for each different dataset? If that is true, it seems to me the generalization of this approach is limited compared to FID, which only uses a ImageNet pre-trained Inception-V3 model.
2. Have the authors try using the proposed FLD and D-FLD to evaluation state-of-the-art image generation models? I am curious to learn if the trend of FID still holds when FLD and D-FLD are used.

---

### Meta-Review · Area_Chair_D3wT · 2024-12-19

**Metareview:**

This paper proposes normalising flow-based evaluation metrics, capable of assessing generated images in terms of how well they align with real image distributions. Evaluation of (generated) image quality can be considered an important problem, with popular existing metrics having noted limitations (e.g. stability, compute requirements).

The authors elected not to submit a rebuttal nor engage with the reviewers during the review period. This left all posed questions unanswered and the majority of reviewers to retain their (in some cases, strongly) negative scores.

The core of the approach can be considered somewhat interesting however, given the above, AC believes that weaknesses outweigh the strengths and therefore recommends rejection.

**Additional Comments On Reviewer Discussion:**

A subset of reviewers agreed that the submission addresses an important and topical problem and that the paper is easy to follow. All reviewers raised concerns and further questions on various important aspects, including: generalisation, efficiency, robustness, reliability, consistency of the narrative, experimental methodology and completeness.

The authors elected not to submit a rebuttal nor engage with the reviewers during the review period. This left all posed questions unanswered and the majority of reviewers to retain their (in some cases, strongly) negative scores.

---

### Decision · Program_Chairs · 2025-01-22

Reject